# Marker for kidney fibrosis is associated with inflammation and deterioration of kidney function in people with type 2 diabetes and microalbuminuria

**Christina Gjerlev Poulsen**[1]*, **Daniel G. K. Rasmussen**[2], **Federica Genovese**[2], **Tine W. Hansen**[1], **Signe Holm Nielsen**[2], **Henrik Reinhard**[1], **Bernt Johan von Scholten**[1,3], **Peter K. Jacobsen**[4], **Hans-Henrik Parving**[5], **Morten Asser Karsdal**[2], **Peter Rossing**[1,6], **Marie Frimodt-Møller**[1]

**1** Steno Diabetes Center Copenhagen, Herlev, Denmark, **2** Nordic Bioscience, Herlev, Denmark, **3** Novo Nordisk A/S, Søborg, Denmark, **4** Department of Cardiology, Rigshospitalet, University of Copenhagen, Copenhagen, Denmark, **5** Department of Endocrinology, Rigshospitalet, University of Copenhagen, Copenhagen, Denmark, **6** Department of Clinical Medicine, University of Copenhagen, Copenhagen, Denmark

* christina.gjerlev.poulsen@regionh.dk

**Data Availability Statement:** Data cannot be shared publicly because potentially identifying and sensitive patient information. Data are available

## Abstract

### Background

Diabetic kidney disease is a major cause of morbidity and mortality. Dysregulated turnover of collagen type III is associated with development of kidney fibrosis. We investigated whether a degradation product of collagen type III (C3M) was a risk marker for progression of chronic kidney disease (CKD), occurrence of cardiovascular disease (CVD), and mortality during follow up in people with type 2 diabetes (T2D) and microalbuminuria. Moreover, we investigated whether C3M was correlated with markers of inflammation and endothelial dysfunction at baseline.

### Methods

C3M was measured in serum (sC3M) and urine (uC3M) in 200 participants with T2D and microalbuminuria included in an observational, prospective study at Steno Diabetes Center Copenhagen in Denmark from 2007–2008. Baseline measurements included 12 markers of sssinflammation and endothelial dysfunction. The endpoints were CVD, mortality, and CKD progression (>30% decline in eGFR).

### Results

Mean (SD) age was 59 (9) years, eGFR 90 (17) ml/min/1.73m$^2$ and median (IQR) urine albumin excretion rate 102 (39–229) mg/24-h. At baseline all markers for inflammation were positively correlated with sC3M (p≤0.034). Some, but not all, markers for endothelial dysfunction were correlated with C3M. Median follow-up ranged from 4.9 to 6.3 years. Higher sC3M was associated with CKD progression (with mortality as competing risk) with a hazard

from the Research & Innovation Office at Steno Diabetes Center Copenhagen (contact via SDC-FP-RIO@regionh.dk) for researchers who meet the criteria for access to confidential data.

**Funding:** DGKR received funding from Danish Research Foundation (https://dg.dk/en/). SHN received funding from Innovation Fund Denmark (https://innovationsfonden.dk/en). The funders had no role in study design, data collection and analysis, decision to publish, or preparation of the manuscript.

**Competing interests:** I have read the journal's policy and the authors of this manuscript have the following competing interests: DGKR, SHN, FG, and MAK are employees of Nordic Bioscience. Nordic Bioscience is a privately owned, small- to medium-sized enterprise, partly focused on the development of biomarkers. None of the authors received fees, bonuses, or other benefits for the work regarding this article. DGKR, SHN, FG, and MAK holds stocks in Nordic Bioscience. The funder provided support in the form of salaries for DGKR and SHN but did not play any additional role in the study design, data collection and analysis, decision to publish, or preparation of the manuscript. Outside this work, BJvS is employed by Novo Nordisk, and PR has received institutional grants from Bayer, Novo Nordisk and AstraZeneca and has acted as consultant for Novo Nordisk, Bayer, Astellas, Boehringer Ingelheim, AstraZeneca, Gilead, Merk, Mundipharma, and Sanofi (honoraria to institution). MFM has received lecture fees from Novartis, Sanofi, Boehringer Ingelheim and Baxter. This does not alter our adherence to PLOS ONE policies on sharing data and materials.

ratio (per doubling) of 2.98 (95% CI: 1.41–6.26; p = 0.004) adjusted for traditional risk factors. uC3M was not associated with CKD progression. Neither sC3M or uC3M were associated with risk of CVD or mortality.

## Conclusions

Higher sC3M was a risk factor for chronic kidney disease progression and was correlated with markers of inflammation.

## Introduction

Diabetic kidney disease (DKD) is a major cause of morbidity and mortality in diabetes. It is the primary cause of end stage kidney disease (ESKD) in western countries and causes up to half of incident cases [1]. However, the majority never reach ESKD because they are more likely to die of cardiovascular disease (CVD). The risk of CVD increases almost exponentially as kidney function declines [2–4].

Regardless of the etiology, a main feature of the progression of chronic kidney disease (CKD) is a pathological deposition of extracellular matrix components, which can trigger renal fibrosis and lead to ESKD [5]. The main structural component of the fibrotic core is collagen and one of the most prominent collagens in the fibrotic kidney is collagen type III. C3M is a degradation product of collagen type III, produced by the matrix metalloproteinase (MMP)-9. C3M thereby reflects turnover of collagen type III in the interstitial matrix and can be considered as a marker for fibrotic activity [6]. Studies have shown increased MMP-9 activity in DKD [7], and increased levels of MMP-9 in plasma were a risk factor for development of microalbuminuria in persons with type 2 diabetes (T2D) [8]. Increased levels of C3M measured in urine has been associated with severity of CKD in persons with type 1 diabetes (T1D) [9], and with both severity and progression of the disease in other CKD cohorts [6, 10]. C3M has not yet been investigated in people with type 2 diabetes and diabetic kidney disease.

Endothelial dysfunction and inflammation play an important role in the onset and progression of fibrosis. In previously reported data in this study population, markers for endothelial dysfunction and inflammation were independently associated with CVD and all-cause mortality [11].

A kidney biopsy is the only current method for detecting renal fibrosis. As fibrosis may be present before clinically detectable kidney disease, fibrotic biomarkers could potentially be used as a non-invasive method for much earlier detection of disease. Additionally, fibrotic biomarkers could be used for disease monitoring and assessment of treatment response.

In this study, we investigated whether C3M, measured in serum and urine, was associated with markers of inflammation and endothelial dysfunction at baseline, and whether it was a risk marker for progression of chronic kidney disease, occurrence of CVD events, and mortality during follow-up in people with T2D and microalbuminuria.

## Materials and methods

### Participants and study procedures

In 2007–2008 we recruited 200 persons with T2D from the outpatient clinic at Steno Diabetes Center Copenhagen, Denmark to a prospective, observational follow-up study. The enrolment criteria, as previously described [12, 13], included: a diagnosis of T2D according to the WHO

criteria, persistent urine albumin excretion rate (UAER) >30 mg/24h (in two out of three consecutive measurements), normal kidney function and no coronary heart disease. All participants received intensive multifactorial treatment consisting of glycemic, lipid and blood pressure control, antithrombotic therapy, and lifestyle intervention according to the Steno 2 study [14]. The study was approved by the local ethics committee (*De Videnskabsetiske Komitéer Region Hovedstaden*) and complied with the Declaration of Helsinki. All participants provided written informed consent.

## Measurement of C3M in serum and urine

The biomarker C3M was measured in serum (sC3M) and urine (uC3M) from samples collected at baseline and were available in 198 and 190 of the 200 participants, respectively. Samples were stored at -80˚C until analyses. Both serum and urine C3M were measured using competitive enzyme-linked immunosorbent assays (ELISAs) developed by Nordic Bioscience, Denmark [10, 15]. MMP-9 mediated degradation of collagen type III, produces a 10 amino acid neo-epitope (610'.KNGETGPQGP'619) (Fig 1). ELISAs were performed using two different monoclonal antibodies for detecting C3M in serum and urine, both antibodies specifically detected all fragments entailing the neo-epitope [15]. Intra- and inter-assay variations of the ELISAs were below 10 and 15%, respectively. The assays were carried out as previously described [10, 15]. To normalize for urine output, uC3M levels were divided by urinary creatinine levels measured on ADVIA 1800 Chemistry System (Siemens Healthineers, Germany).

## Baseline clinical and laboratory measures

UAER was measured in three 24h urine collections by an enzyme immunoassay (Vitros, Raritan, NJ, USA). HbA1c, plasma creatinine and serum cholesterol were determined using standard methods. The estimated glomerular filtration rate (eGFR) was calculated applying the Chronic Kidney Disease Epidemiology Collaboration (CKD-EPI) equation [16]. Current smoking was defined as one or more cigarettes, cigars, or pipes per day. Brachial blood pressure was the average of two consecutive measurements after 10 minutes of rest.

Markers of inflammation (TNF-α, sICAM-1, sICAM-3, hsCRP, SAA, IL-1beta, IL-6 and IL-8) and endothelial dysfunction (thrombomodulin, sVCAM-1, sE-selectin and sP-selectin) were measured at baseline using MSD multipanel measurements or ELISA as previously described [17].

## Follow-up

Approximately annual measurements of plasma creatinine were performed in 175 of the 198 participants (88.4%) as part of their regular diabetes control at Steno Diabetes Center Copenhagen. The exact time from baseline to each creatinine measurement varied highly between each individual and each measurement. Information on participants with available samples was traced through the Danish National Death and Danish National Health Registries on January 1, 2014. No participants were lost to follow-up. Definitions of the three predefined endpoints have previously been described [13, 18]. The combined cardiovascular endpoint included cardiovascular mortality, non-fatal myocardial infarction, stroke, ischemic CVD, and heart failure. In case of multiple events in one participant, only the first was included. Chronic kidney disease progression was defined as a >30% decline in eGFR based on approximately annual plasma creatinine measurements (five in total) and evaluated as change from baseline to the last available measurement.

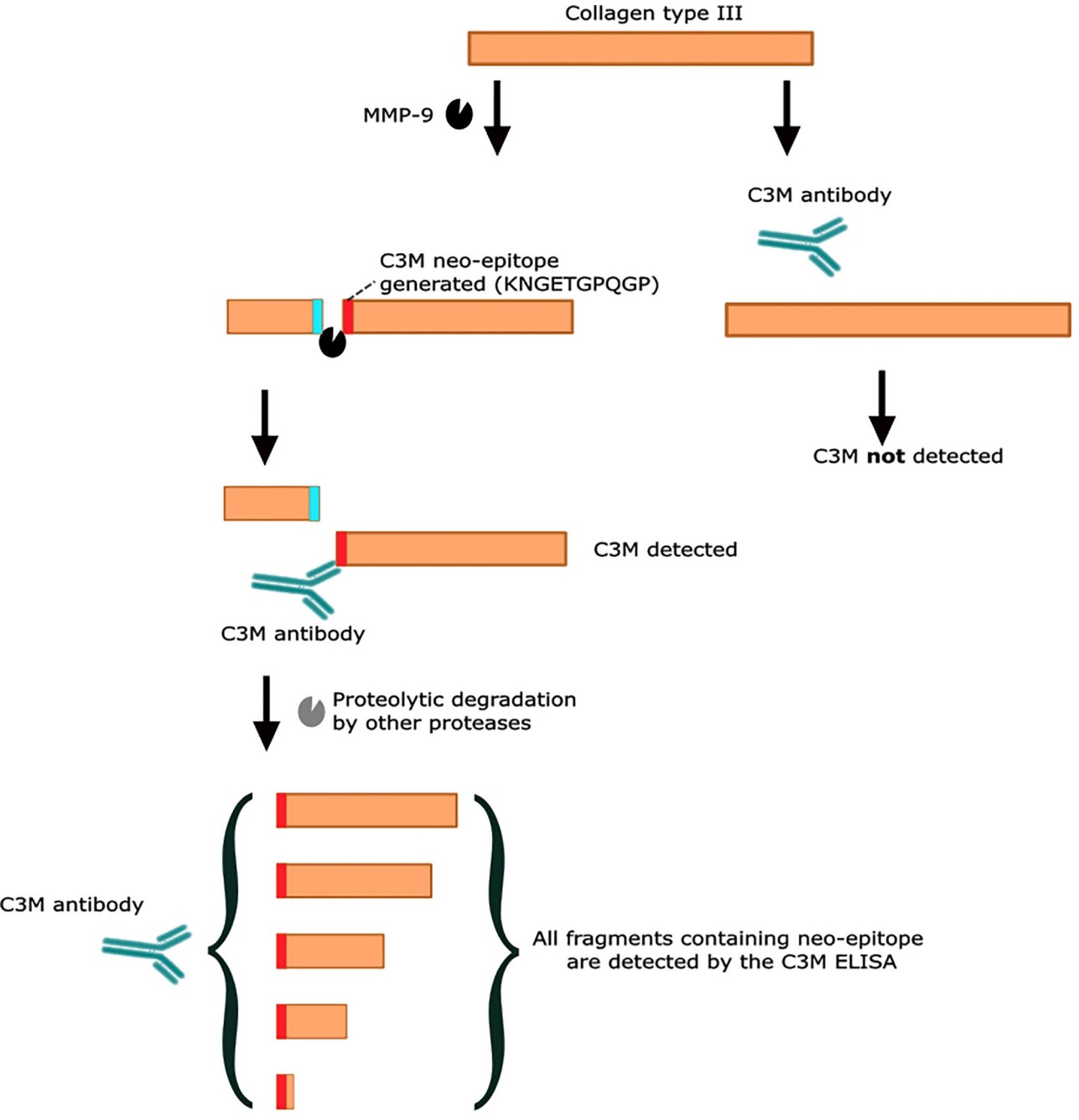

**Fig 1. Generation and detection of C3M fragments.** C3M is generated by degradation of collagen type III by the matrix metalloproteinase (MMP)-9, which produces a 10 amino acid neo-epitope (KNGETGPQGP). The ELISA antibody detects this neo-epitope and can thereby detect fragments of down to 10 amino acids. ELISA: enzyme-linked immunosorbent assay.

### Statistical analyses

Normally distributed variables are expressed as mean values ± standard deviations (SD). Both sC3M and uC3M had skewed distributions (S1 Fig, panel A and B) and were log2-transformed in all analyses and presented as medians with interquartile range (IQR). Categorical variables

are presented as total numbers with corresponding percentages. Clinical characteristics were stratified by tertiles of the concentrations of sC3M and compared using one-way ANOVA for normally distributed variables, Kruskal-Wallis test for skewed distributed variables and the $\chi^2$-test for categorical variables.

Cox proportional hazards analysis was used to calculate hazard ratios (HR) with 95% confidence intervals (CI) per doubling of sC3M and uC3M for the three endpoints and with mortality as competing risk in the analysis of eGFR decline. Adjustment for all models included traditional cardiovascular risk factors including sex, age, body mass index, LDL cholesterol, smoking, HbA1c, plasma creatinine, systolic blood pressure and UAER.

Cox proportional hazard analysis was applied to compare the risk of eGFR decline according to tertiles of sC3M and uC3M, adjusted for traditional cardiovascular risk factors as described.

Linear mixed effect models were applied to examine the association between sC3M/uC3M and eGFR decline during follow-up (including the five follow up measurements). The model was applied using the "gls function" with compound symmetry covariance structure in the "nlme package" using R version 4.1.0.

To evaluate the added predictive value for the markers of inflammation over the value of sC3M, we employed receiver operating characteristic (ROC) curves, applying C-statistics for area under curve (AUC) analysis. For reasons of statistical efficiency, we calculated a z-score for the markers of inflammation by averaging the individual biomarkers signed z-scores. The base model included the traditional cardiovascular risk factors and sC3M, and in the second model we added the z-score for the markers of inflammation.

A two-sided P-value <0.05 was considered statistically significant. Statistical analyses were performed using the SAS software (version 9.4, SAS Institute, Cary, NC, USA).

## Results

### Baseline characteristics

All participants were Caucasian, 149 (75%) were male, mean (SD) age was 58.6 (8.7) years, diabetes duration 12.7 (7.4) years, eGFR was 90 (17) ml/min/1.73m$^2$, and median (IQR) UAER was 102 (39–229) mg/24h. The median (IQR) concentration of sC3M was 8.4 (6.9–10.3) ng/ml and uC3M was 6.09 (4.52–7.68) ng/mol. Most were treated with oral antidiabetic medications (85%) and nearly all received antihypertensive therapy (99%) and statins (94%). Baseline characteristics of both the total population and in tertiles of sC3M are presented in Table 1. There were no significant differences between the tertiles for any of the variables. However, there was a trend towards more women in the highest tertile. Baseline characteristics for tertiles of uC3M are presented in S1 Table. Individuals in the highest tertile of uC3M are more likely to be female, be younger of age, have a shorter diabetes duration, a higher HbA$_{1c}$ and a higher eGFR at baseline.

There was no correlation between sC3M and uC3M ($R^2$ = 0.002; p = 0.56; S1 Fig, panel C) or between sC3M and either eGFR or UAER (p≥ 0.37; S1 Fig, panel D and E). uC3M was positively correlated with eGFR ($R^2$ = 0.09; p < 0.001), but not correlated with UAER ($R^2$ = 0.015; p = 0.09).

### Baseline correlations of C3M and markers of inflammation and endothelial dysfunction

As shown in Table 2, all markers of inflammation were positively correlated with sC3M both in unadjusted and adjusted analyses (p ≤ 0.034). uC3M was positively correlated with IL-6 in

**Table 1. Baseline characteristics of the total population and stratified by sC3M tertiles.**

| Characteristic | All | Tertile 1 | Tertile 2 | Tertile 3 | P-value |
|---|---|---|---|---|---|
| | n = 198 | n = 61 | n = 68 | n = 69 | |
| sC3M (ng/ml) | 8.4 (6.9–10.3) | 6.3 (5.8–6.8) | 8.3 (7.8–8.8) | 11.4 (10.5–13.5) | |
| Male, n (%) | 149 (75) | 49 (80) | 55 (81) | 45 (65) | 0.06 |
| Age (years) | 58.6 ± 8.7 | 59.3 ± 7.7 | 59.0 ± 8.3 | 57.7 ± 10.0 | 0.30 |
| Known duration of diabetes (years) | 12.7 ± 7.4 | 14.2 ± 6.8 | 11.3 ± 6.7 | 12.8 ± 8.3 | 0.30 |
| Body mass index (kg/m2) | 32.5 ± 5.8 | 32.6 ± 5.5 | 31.8 ± 5.5 | 33.1 ± 6.2 | 0.60 |
| HbA$_{1c}$ (%) | 7.9 ± 1.3 | 7.9 ± 1.4 | 7.6 ± 1.2 | 8.0 ± 1.5 | 0.72 |
| HbA$_{1c}$ (mmol/mol) | 62.3 ± 14.8 | 63.1 ± 14.9 | 60.0 ± 12.8 | 63.9 ± 16.3 | 0.72 |
| Urinary albumin excretion rate (mg/24-h) | 102 (39–229) | 99 (37–191) | 122 (47–240) | 81 (40–361) | 0.48 |
| P-creatinine (μmol/L) | 76.4 ± 18.4 | 76.5 ± 20.4 | 75.0 ± 16.7 | 77.6 ± 18.2 | 0.72 |
| eGFR (ml/min/1.73m$^2$) | 90 ± 17 | 90 ± 17 | 91 ± 16 | 88 ± 19 | 0.43 |
| LDL cholesterol (mmol/L) | 1.9 ± 0.8 | 1.8 ± 0.7 | 1.8 ± 0.8 | 2.0 ± 0.9 | 0.29 |
| Systolic blood pressure (mmHg) | 130 ± 17 | 133 ± 18 | 128 ± 16 | 130 ± 17 | 0.53 |
| Diastolic blood pressure (mmHg) | 75 ± 9 | 76 ± 12 | 74 ± 10 | 75 ± 11 | 0.99 |
| Current smoker, n (%) | 59 (30) | 21 (34) | 16 (24) | 22 (32) | 0.36 |
| Treatment with | | | | | |
| Oral antidiabetic, n (%) | 169 (85) | 55 (90) | 59 (87) | 55 (80) | 0.22 |
| Insulin, n (%) | 122 (62) | 38 (62) | 42 (62) | 42 (61) | 0.99 |
| Antihypertensive drugs, n (%) | 196 (99) | 61 (100) | 68 (100) | 67 (97) | 0.15 |
| RAAS blockade, n (%) | 194 (98) | 61 (100) | 67 (99) | 66 (96) | 0.20 |
| Statin, n (%) | 187 (94) | 56 (92) | 67 (99) | 64 (93) | 0.19 |
| Aspirin, n (%) | 182 (92) | 56 (92) | 64 (94) | 62 (90) | 0.66 |

Data are expressed as mean ± SD, median (interquartile range) or number (%) as appropriate. eGFR: estimated glomerular filtration rate. HbA1c: glycosylated hemoglobin; RAAS: renin-angiotensin-aldosterone-system. P-value for trend across tertiles of sC3M.

the adjusted analyses (p = 0.040), but no other correlations between uC3M and markers of inflammation were demonstrated. Thrombomodulin and sVCAM-1 were positively correlated with sC3M in unadjusted analyses, and the correlation for sVCAM-1 remained after adjustment (p < 0.001). Thrombomodulin was negatively correlated with uC3M in unadjusted analyses (P < 0.001), but not after adjustment. sE-selectin was positively correlated with uC3M both in unadjusted and adjusted analyses (p ≤ 0.048). The other markers of endothelial dysfunction were not correlated with neither sC3M nor uC3M.

## Incidence of cardiovascular events, mortality and decline in eGFR

The median (IQR) follow-up time in years was 6.1 (5.9–6.6) for CVD events, 6.3 (6.0–6.7) for mortality, and 4.6 (3.8–5.1) for eGFR decline. During the follow-up period 40 participants reached the combined CVD endpoint, 26 died and 42 declined >30% in eGFR from baseline. The combined CVD endpoint included 11 fatal CVD events (two events of acute myocardial infarction, one case of ischemic CVD, six sudden and otherwise unexplained events and two events of heart failure) and 29 non-fatal CVD events (three cases of acute myocardial infarction, three strokes, 19 cases of ischemic CVD and four cases of heart failure). Out of the 26 participants who died, 11 were due to CVD, nine related to cancer and six related to other causes. No participants were diagnosed with kidney failure during follow-up.

**Table 2. Unadjusted and adjusted correlations between serum and urine C3M and markers of inflammation and endothelial dysfunction.**

| | Serum C3M | | | | Urine C3M | | | |
|---|---|---|---|---|---|---|---|---|
| | Unadjusted | | Adjusted | | Unadjusted | | Adjusted | |
| | β (95%CI) | P-value | β (95%CI) | P-value | β (95%CI) | P-value | β (95%CI) | P-value |
| **Inflammation markers** | | | | | | | | |
| TNF-alfa | 0.15 (0.09–0.21) | <**0.001** | 0.14 (0.07–0.21) | <**0.001** | -0.08 (-0.18–0.03) | 0.15 | -0.04 (-0.14–0.07) | 0.46 |
| sICAM-1* | 0.15 (0.09–0.21) | <**0.001** | 0.14 (0.08–0.21) | <**0.001** | 0.07 (-0.04–0.17) | 0.20 | 0.03 (-0.07–0.13) | 0.53 |
| sICAM-3* | 0.08 (0.02–0.14) | **0.008** | 0.07 (0.01–0.13) | **0.034** | 0.07 (-0.04–0.17) | 0.22 | 0.06 (-0.04–0.16) | 0.23 |
| hsCRP | 0.18 (0.13–0.24) | <**0.001** | 0.20 (0.14–0.71) | <**0.001** | 0.07 (-0.03–0.18) | 0.15 | 0.06 (-0.04–0.16) | 0.25 |
| SAA | 0.15 (0.09–0.21) | <**0.001** | 0.15 (0.09–0.21) | <**0.001** | 0.05 (-0.05–0.15) | 0.33 | 0.05 (-0.05–0.14) | 0.36 |
| IL-1beta | 0.09 (0.03–0.15) | **0.005** | 0.10 (0.02–0.15) | **0.007** | 0.02 (-0.09–0.13) | 0.74 | 0.04 (-0.05–0.14) | 0.38 |
| IL-6 | 0.13 (0.07–0.19) | <**0.001** | 0.14 (0.07–0.20) | <**0.001** | 0.05 (-0.06–0.15) | 0.39 | 0.10 (0.01–0.20) | **0.040** |
| IL-8 | 0.11 (0.04–0.17) | <**0.001** | 0.11 (0.04–0.17) | **0.013** | -0.04 (-0.14–0.06) | 0.44 | -0.009 (-0.10–0.09) | 0.84 |
| **Markers of endothelial dysfunction** | | | | | | | | |
| Thrombomodulin | 0.07 (0.01–0.14) | **0.019** | 0.07 (-0.01–0.14) | 0.074 | -0.17 (-0.27- -0.07) | <**0.001** | -0.10 (-0.21–0.01) | 0.078 |
| sVCAM-1 | 0.15 (0.09–0.21) | <**0.001** | 0.15 (0.08–0.21) | <**0.001** | 0.04 (-0.06–0.15) | 0.41 | 0.03 (-0.05–0.16) | 0.53 |
| sE-selectin | -0.01 (-0.07–0.05) | 0.71 | -0.02 (-0.08–0.06) | 0.74 | 0.10 (0.01–0.21) | **0.048** | 0.10 (0.01–0.20) | **0.040** |
| sP-selectin | -0.01 (-0.08–0.04) | 0.49 | -0.02 (-0.09–0.06) | 0.64 | 0.01(-0.09–0.11) | 0.84 | 0.014 (-0.09–0.11) | 0.79 |

The β -estimates with 95% confidence interval (CI) represent standardized effect (log2 transformed). Adjustment included sex, age, body mass index, LDL cholesterol, smoking, HbA1c, plasma creatinine, systolic blood pressure and urinary albumin excretion rate.

*Also markers of endothelial dysfunction. Bold indicate a significant p-value.

## C3M as risk marker

Table 3 shows the associations between sC3M, uC3M and the three endpoints. High sC3M was associated with an increased risk of a >30% decline in eGFR in the unadjusted and adjusted analyses, including mortality as competing risk. Serum C3M was not a risk marker

**Table 3. Unadjusted and adjusted associations between serum and urine C3M and development of cardiovascular events, all-cause mortality and decline in eGFR.**

| | Cardiovascular events | | All-cause mortality | | Decline in eGFR > 30% with death as competing risk | |
|---|---|---|---|---|---|---|
| | (n = 40) | | (n = 26) | | (n = 42)* | |
| | HR (95% CI) | P-value | HR (95% CI) | P-value | HR (95% CI) | P-value |
| **Serum C3M** | | | | | | |
| Unadjusted | 1.17 (0.57–2.40) | 0.67 | 1.57 (0.67–3.71) | 0.30 | **2.71 (1.38–5.30)** | **0.004** |
| Adjusted | 1.52 (0.67–3.42) | 0.31 | 1.80 (0.69–4.73) | 0.23 | **2.98 (1.41–6.26)** | **0.004** |
| **Urine C3M** | | | | | | |
| Unadjusted | 0.76 (0.51–1.12) | 0.16 | 0.88 (0.52–1.49) | 0.21 | 0.94 (0.70–1.28) | 0.70 |
| Adjusted | 0.83 (0.60–1.14) | 0.25 | 0.93 (0.62–1.40) | 0.74 | 0.96 (0.66–1.38) | 0.82 |

Cox regression analyses of C3M in relation to risk of fatal and nonfatal cardiovascular events, all-cause mortality and decline in eGFR >30% in 198 participants, number of events given for each endpoint. Values are hazard ratios (HR) with 95% confidence intervals (CI) per doubling of C3M. Adjustment included sex, age, body mass index, LDL cholesterol, smoking, HbA1c, plasma creatinine, systolic blood pressure and urinary albumin excretion rate.

*Data on decline in eGFR was only available for 175 out of 198 participants (88.4%).

eGFR: estimated glomerular filtration rate.

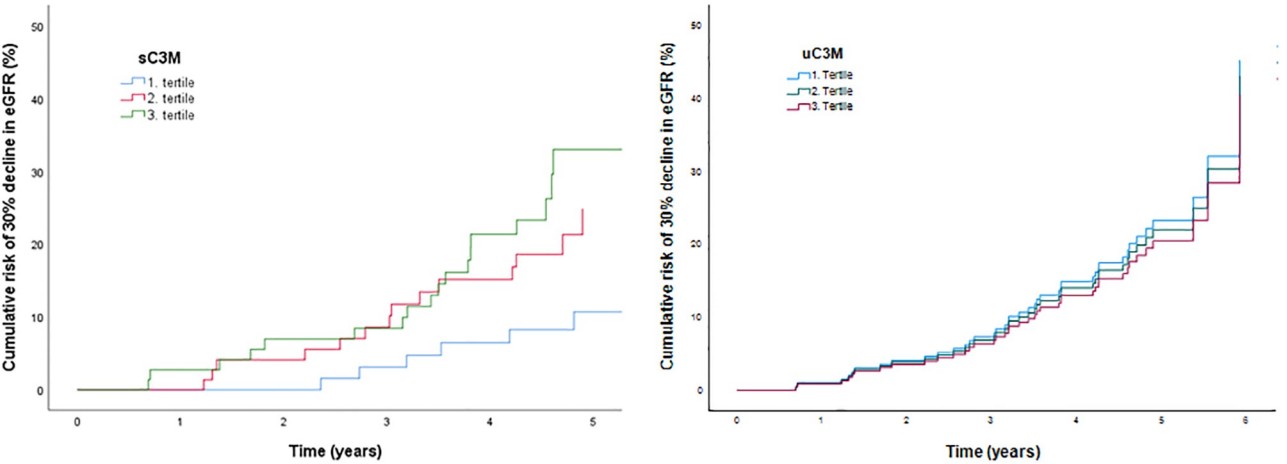

**Fig 2. Cox survival plot of decline in eGFR >3 0% for sC3M and uC3M.** Adjustment included sex, age, body mass index, LDL cholesterol, smoking, HbA$_{1c}$, plasma creatinine, systolic blood pressure and urinary albumin excretion rate. uC3M: urinary C3M, sC3M: serum C3M.

for CVD events or all-cause mortality, while uC3M was not associated with any of the endpoints.

Fig 2 shows the Cox proportional hazard plots with tertiles of sC3M/uC3M and a >30% decline in eGFR. High levels of sC3M were associated with a significantly increased risk of decline in eGFR > 30% (p = 0.047). The risk was significantly higher in tertile 3 compared with tertile 1 (p = 0.014). Tertile 2 was not significantly different from tertile 1 (p = 0.132). There was no association between tertiles of uC3M and development of a >30% decline in eGFR (p = 0.96).

The mixed model analysis showed a significant association between sC3M and eGFR decline over time in both the unadjusted (p = 0.037) and adjusted model (p = 0.048), including adjustment for sex, age, body mass index, LDL cholesterol, smoking, HbA1c, plasma creatinine, systolic blood pressure and urinary albumin excretion rate. Urine C3M was not associated with eGFR decline over time in neither the unadjusted (p = 0.24) or adjusted model (p = 0.36).

## Additional analyses

Because of the strong correlation between sC3M and markers of inflammation, we performed additional adjustment for the markers of inflammation for all the outcomes. Results were confirmatory and higher sC3M remained associated with an increased risk of a >30% decline in eGFR (p = 0.001).

Adding the z-score for the inflammation markers to ROC-curves with sC3M and traditional cardiovascular risk factors did not improve the predictive value on any of the endpoints (CVD events: p = 0.36; all-cause mortality: p = 0.32; >30% decline in eGFR: p = 0.75).

Urinary C3M were divided with urine creatinine to normalize for urine output. Since this is correlated with the serum creatinine level, we performed a sensitivity analysis not including serum creatinine in the adjusted model for eGFR decline. This did not change the estimate or the lack of association with eGFR decline, HR (95% CI): 0.88 (0.63–1.25), p-value 0.47.

Additional sensitivity analysis showed that adding antihypertensive treatment to the model of eGFR decline did not change the results, HR (95% CI): 2.94 (1.40–6.19).

## Discussion

In the present study, we demonstrated that sC3M was positively correlated with markers of inflammation and endothelial dysfunction at baseline. Moreover, higher sC3M was a risk marker for progression of kidney disease during follow-up. In contrast, sC3M was not a risk marker for development of CVD or mortality and uC3M was not a risk marker for any of the endpoints.

We found no correlation between serum and urine C3M. This phenomenon has been observed across several other studies [6, 9]. One explanation could be that C3M fragments have different lengths and that a large proportion of the circulating fragments are not freely filtered into the urine. Another explanation could be that C3M produced by collagen type III degradation in the renal stoma were released directly into the urine and thereby not systemically available.

### Baseline correlations between C3M and markers of inflammation and endothelial dysfunction

Endothelial dysfunction and chronic inflammation are considered important factors in the pathogenesis of DKD. Previous studies have found markers of both endothelial dysfunction and inflammation associated with onset and progression of albuminuria [19, 20], with decline in eGFR [21] and with development of CVD and mortality [17].

Our results showed consistent, positive correlations between sC3M and markers of inflammation and some of the markers of endothelial dysfunction. Supporting these findings, another study showed a positive correlation between sC3M and CRP in CKD [6]. These consistent correlations indicate that these pathways are interconnected in the pathogenesis of DKD. Despite the consistent and strong correlations, sC3M was not associated with development of CVD and mortality in this population, even though associations between the markers of inflammation and endothelial dysfunction and risk of both CVD and mortality have previously been shown in the same population [17]. This adds evidence to the conclusion that sC3M is related to kidney disease but not CVD. The findings from the ROC analyses showed that adding the markers of inflammation to a model including traditional cardiovascular risk factors and sC3M did not add predictive value for any of the endpoints.

In contrast to sC3M, uC3M was only correlated with IL-6 in adjusted analyses, and not correlated with any of the other markers of inflammation and was positively correlated with only one of the markers of endothelial dysfunction.

### Serum C3M as a risk marker

In a study in persons with T1D, Pilemann-Lyberg et al. found higher sC3M to be a risk marker for a >30% decline in eGFR and for development of ESKD in the unadjusted models, but the association was lost after adjustment [9]. Similar findings were demonstrated in a population with CKD, where higher sC3M was associated with higher risk of CKD progression, but the significance was lost in the adjusted analysis [6].

None of the clinical characteristics, including kidney function and albuminuria, differed between the tertiles of sC3M at baseline. Since sC3M was a risk marker for kidney disease progression, this could mean that sC3M, has a value as risk marker, that are not evident from traditional measures of kidney function.

Consistently with previous findings, sC3M was not a risk marker for CVD or mortality in this population. In T1D, Pilemann-Lyberg et al found an association between higher sC3M and risk of CVD and mortality, but the significance was lost after adjustment for risk markers

[9]. Similarly, in the study by Genovese et al, sC3M was not a risk marker for mortality in the adjusted model [6].

Serum C3M is highly associated with markers of systemic inflammation. But despite the evident association between systemic inflammation and both CVD, mortality and kidney disease progression, sC3M seems only to be associated with kidney disease progression.

## Urine C3M as a risk *marker*

In the population with T1D, the authors found associations between lower uC3M and higher risk of a >30% decline in eGFR and development of ESKD in unadjusted models, but associations were lost after adjustment [9]. In the present study we saw the same inverse association, but statistically insignificant. Genovese et al showed an association between higher levels of uC3M and lower risk of CKD progression in a mixed CKD population. A sub-analysis confirmed the findings in a diabetes subpopulation (diabetes type was not specified), however the participants were not necessarily diagnosed with DKD [6]. All mentioned uC3M measurements have been adjusted for urine output by dividing with urine creatinine concentration. The potential of uC3M as a risk marker in CKD patients seems to depend on the investigated population, and in the population investigated in this work, which includes type 2 diabetes patients with early to moderate kidney disease, the uC3M marker was not informative. The lack of correlation with markers for inflammation and endothelial dysfunction supports that the urinary level of C3M is not a valuable measure in this population. Further investigations are required in patients with more advanced DKD to establish whether this can be a risk marker in a subpopulation of DKD.

## Strengths and limitations

A strength of the study was the prospective design and the adequate length of follow-up. Participants were consecutively recruited which minimized selection bias and there was a limited loss to follow-up and missing data. The cohort was well-characterized, which enabled adjustment for important risk factors.

The relatively small study population and thereby a low number of events was a limitation. The cohort consisted of persons with T2D and microalbuminuria without known CVD, which limits the generalizability of the results. Comorbidities could have influenced the results, as fibrotic activity in other organs might affect levels of C3M. Likewise levels of uC3M can be affected by impaired excretion following declining kidney function and can be altered by non-selective proteinuria in kidney diseases like DKD [22].

Future studies should elaborate on the role of C3M as a biomarker for predicting kidney disease progression and the potential effects of therapeutic modification. In rodents, sodium-glucose transport protein 2 (SGLT2) inhibitors have shown to have anti-inflammatory and antifibrotic effects [23, 24]. Recently the nonsteroidal mineralocorticoid receptor antagonist finerenone was demonstrated to reduce progression of CKD and development of CVD (particularly reduced hospitalization for heart failure) in two large trials FIDELIO-DKD and FIGARO-DKD including patients with type 2 diabetes and a broad range of CKD with UACR >30 mg/g and eGFR>25 ml/min/1.73m$^2$ [25, 26]. Experimental studies have demonstrated that blocking mineralocorticoid receptor overactivation protects the kidney by reduced inflammation and fibrosis without affecting blood pressure [27]. Decreasing C3M levels with finerenone could be a way to suggest the same mode of action in humans. Unpublished results have shown that treatment with the glucagon-like peptide-1 receptor agonist (GLP1RA) dulaglutide in T2D was associated with increasing levels of urinary C3M. This could potentially explain some of the beneficial treatment effects of both SGLT2 inhibitors and GLP1RAs.

In conclusion, we found a positive correlation between sC3M and markers of inflammation and endothelial dysfunction, and that elevated sC3M was a risk marker for progression of DKD in persons with type 2 diabetes.

## Supporting information

**S1 Fig. Distribution and correlations.** (A) Distribution of sC3M (B) Distribution of uC3M (C) correlation analysis between sC3M and uC3M (D) correlation analysis between eGFR and sC3M and (E) correlation analysis between sC3M and UAER.
(DOCX)

**S1 Table. Baseline characteristics for urine C3M tertiles.** Data are expressed as mean ± SD, median (interquartile range) or number (%) as appropriate. eGFR: estimated glomerular filtration rate. HbA$_{1c}$: glycosylated hemoglobin; RAAS: renin-angiotensin-aldosterone-system. P-value for trend across tertiles of uC3M.
(DOCX)

## Acknowledgments

Authors want to thank all participants and acknowledge the work of lab technicians Ulla M. Smidt, Berit R. Jensen, Tina R. Juhl and Anne G. Lundgaard, employees at Steno Diabetes Center Copenhagen and MD Viktor Rotbain Curovic for figure preparation.

## Author Contributions

**Conceptualization:** Daniel G. K. Rasmussen, Federica Genovese, Tine W. Hansen, Signe Holm Nielsen, Henrik Reinhard, Peter K. Jacobsen, Hans-Henrik Parving, Peter Rossing, Marie Frimodt-Møller.

**Data curation:** Tine W. Hansen, Henrik Reinhard, Bernt Johan von Scholten.

**Formal analysis:** Christina Gjerlev Poulsen, Daniel G. K. Rasmussen, Federica Genovese, Tine W. Hansen, Signe Holm Nielsen, Peter Rossing, Marie Frimodt-Møller.

**Supervision:** Peter Rossing.

**Visualization:** Tine W. Hansen.

**Writing – original draft:** Christina Gjerlev Poulsen.

**Writing – review & editing:** Christina Gjerlev Poulsen, Daniel G. K. Rasmussen, Federica Genovese, Tine W. Hansen, Signe Holm Nielsen, Henrik Reinhard, Bernt Johan von Scholten, Peter K. Jacobsen, Hans-Henrik Parving, Morten Asser Karsdal, Peter Rossing, Marie Frimodt-Møller.

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
