## [Decision Letter · Decision Letter 0]

21 Dec 2022

PONE-D-22-32292Marker for kidney fibrosis is associated with inflammation and deterioration of kidney function in people with type 2 diabetes and microalbuminuriaPLOS ONE

Dear Dr. Poulsen,

Thank you for submitting your manuscript to PLOS ONE. After careful consideration, we feel that it has merit but does not fully meet PLOS ONE’s publication criteria as it currently stands. Therefore, we invite you to submit a revised version of the manuscript that addresses the points raised during the review process.

We look forward to receiving your revised manuscript.

Kind regards,

Harald Mischak

Academic Editor

PLOS ONE

“I have read the journal's policy and the authors of this manuscript have the following competing interests:

DGKR, SHN, FG, and MAK are employees of Nordic Bioscience. Nordic Bioscience is a privately owned, small- to medium-sized enterprise, partly focused on the development of biomarkers. None of the authors received fees, bonuses, or other benefits for the work regarding this article. DGKR, SHN, FG, and MAK holds stocks in Nordic Bioscience. The funder provided support in the form of salaries for DGKR and SHN but did not play any additional role in the study design, data collection and analysis, decision to publish, or preparation of the manuscript. Outside this work, BJvS is employed by Novo Nordisk, and PR has received institutional grants from Bayer, Novo Nordisk and AstraZeneca and has acted as consultant for Novo Nordisk, Bayer, Astellas, Boehringer Ingelheim, AstraZeneca, Gilead, Merk, Mundipharma, and Sanofi (honoraria to institution). MFM has received lecture fees from Novartis, Sanofi, Boehringer Ingelheim and Baxter.”

Additional Editor Comments:

As you can see from the comments, both reviewers were positive about the study, but raised several issues that should result in an improvement of the manuscript, once adressed. The comments are very constructive, I agree with them and hope that you will be able to fully address them. I would appreciate a statement in the paper clearly indicating that you would be happy to share all data, as required by the journal, but unfortunately are prevented from doing so by the restrictive Danish laws that you are forced to follow (or a similar statement).

Reviewers' comments:

Reviewer's Responses to Questions

**Comments to the Author**

1. Is the manuscript technically sound, and do the data support the conclusions?

Reviewer #1: Yes

Reviewer #2: Partly

2. Has the statistical analysis been performed appropriately and rigorously? 

Reviewer #1: Yes

Reviewer #2: Yes

3. Have the authors made all data underlying the findings in their manuscript fully available?

Reviewer #1: Yes

Reviewer #2: Yes

4. Is the manuscript presented in an intelligible fashion and written in standard English?

Reviewer #1: Yes

Reviewer #2: Yes

5. Review Comments to the Author

Reviewer #1: The study by Poulsen et al., focusses on the association between a circulating fibrosis marker and progression of T2D kidney disease and the inflammatory state of these T2D patients.

This is an interesting study but that would benefit from some improvements.

- As the reader might not be familiar with the transformation of collagen 3 into C3M and what is exactly measured with the ELISA, a figure explaining this would be very helpful (the size of C3M fragment measured, see comment below, is not clear as well).

- The definition of CKD progression was based on >30% decline of eGFR compared to baseline. It would be desirable that this >30% decline was based on repeated creatinine measurements, or is this already the case?

-The uC3M story is not fully clear. uC3M is not mentioned in Table 1, but it is detected in urine as shown in Suppl figure 1. Since C3M is apparently a cleavage fragment and stable (since detected in serum) one would expect a circulating peptide to be filtered by the kidney and then found in urine. Do the authors have any explanation here?

- The last section of the results very briefly describes the interaction of sC3M with the inflammation markers. Since there is a quite strong correlation with those inflammatory markers and sC3M, what about the predictive value on CKD progression of CVD of those individual markers? It would be very interesting to compare the predictive value of those individual markers with sC3M. This is especially important since the authors claim that sC3M would be a potential marker of progression CKD and therefore its added value should be shown and compared to those inflammation markers.

- Related to this, even if after adjustment with inflammation markers sC3M remained associated to outcome, is combination of one or more of those markers (logistic regression, SVMs, random Forrest, etc…) improving the predictive value of progression to CKD or CVD events over sC3M alone?

- The strong association to “endothelial inflammation” of sC3M and the absence to predict CVD is surprising, this is not discussed.

Reviewer #2: Thank you for giving me the opportunity to review this research manuscript. The authors aimed to study the association of serum and urinary C3M (a matrix metalloproteinase-9-mediated degradation product of collagen type III) with progression of chronic kidney disease. They additionally examined the association of markers of systemic inflammation and endothelial dysfunction in relation to C3M. The authors documented that chronic kidney disease progression associates with high levels of serum C3M – urinary C3M was not associated. Please, find below some important statistical and methodological considerations:

Major comments:

- The median follow-up was six years and the creatinine was collected annually, which means patients would have an average of six creatinine measures. Considering this, using a categorical variable excludes important information about the progression of chronic kidney disease. From a clinical perspective, I understand that it makes more sense to show Cox proportional modeling because readers are more familiar with it. Nevertheless, the authors should include the rich information they have about the progression of kidney disease based on repeated creatinine measures. Therefore, I would strongly recommend they conduct and report mixed models. Just as an example, by running mixed modeling, authors could express the following estimates: kidney function decrease -0.2 mg/dL per 2-standard deviation (SD) in the sC3M. Moreover, if they estimate the difference change divided by follow-up time (progression rate), they could express the following: kidney function decrease -0.2 mg/dL per year per each 2-SD increment in the sC3M... It has a different interpretation (estimates) than Cox proportional models, but still, those estimates are clinically meaningful and from a statistical perspective, considering the outcome as a continuous repeated variable provides more accurate information than dividing into a categorical outcome.

- I believe the authors should focus the manuscript on the progression of kidney disease and cardiovascular (CV) diseases as outcomes should be considered as competing risk. The CV outcomes are not part of the authors' hypothesis (which seems they are trying to test different hypotheses making the manuscript more difficult to digest). As they wrote in their introduction about why investigating CV outcomes, this is a pure statistical issue called "competing risk". As this is a predictive study, authors should treat CV disease and mortality as potential competing risks for the progression of kidney disease. That is, because these are high-risk patients, they can potentially die or suffer CV events (competing risk) before they develop the event (progression of kidney disease). Please, see the following paper which might help the authors to address their hypothesis and redefine their statistical approach: “Noordzij M. et al. (2013). When do we need competing risks methods for survival analysis in nephrology?. Nephrology Dialysis Transplantation, 28(11), 2670-2677.” Therefore, they should document the sub-hazard ratios for progression of kidney disease by adjusting models for total mortality and CV deaths as competing risk.

- If the authors divided uC3M levels by urinary creatinine levels, and then estimated glomerular filtration rate (GRF) based on serum creatinine levels, should not the division affect the authors analysis? I understand the biological reasoning, but there is a statistical consideration about this. Can the authors perform the analysis for uC3M without this adjustment/standardization by serum creatinine levels?

- Figure 1, can the authors indicate the cutoff points for the t1, t2, and t3? Moreover, authors should standardize figure 1 by age (tertile groups, or two groups based on the mean age) and sex groups. Another alternative is to standardized the incidence curve for the average population of con founders (see example). Otherwise, these are crude analysis. Can the authors also provide the same curve for uC3M? I understand that it would not be significant, but the utility of Figure one does not only rely on reporting the absolute risk for progression of chronic kidney disease but also to facilitate readers quick results of the study.

Minor comments:

- Can the authors specify how much is the SD doubling in the log-transformed C3M?

- Table 2, can the authors provide the 95% CI?

- There is no documentation of how urinary C3M was measured. Similar to sC3M?

- Change title of table 3.

- There is an extra decimal in the last estimate of table 3.

- Why not include BP treatment as confounder.

- Specify follow-up in time in the abstract.

- How many creatinine measures were collected?

- Supplemental figure 1 can be placed in the manuscript. I believe it is a main (additional) finding that sC3M and uC3M were not correlated. Moreover, it is just one item as supplemental information [it can also be described in the results section as: “sC3M and uC3M were not correlated (P value >x.xxx, data not shown) ….]. On the other hand, a potential supplemental table is to replicate table 1 but for uC3M.

6. PLOS authors have the option to publish the peer review history of their article (what does this mean?). If published, this will include your full peer review and any attached files.

Reviewer #1: No

Reviewer #2: No

---

## [Author Response · Author response to Decision Letter 0]

17 Feb 2023

Point-by-point responses to the Reviewers’ comments. Responses are marked with blue text. 

Reviewer #1: 

The study by Poulsen et al., focusses on the association between a circulating fibrosis marker and progression of T2D kidney disease and the inflammatory state of these T2D patients.

This is an interesting study but that would benefit from some improvements.

- As the reader might not be familiar with the transformation of collagen 3 into C3M and what is exactly measured with the ELISA, a figure explaining this would be very helpful (the size of C3M fragment measured, see comment below, is not clear as well).

We agree that the formation of C3M and the ELISA measurements were not adequately described. We have improved the description in the Introduction and the Methods section and added a figure (Fig 1) picturing the degradation of collagen type III and the ELISA antibody. 

Revised Introduction (page 3, line 9-13): 

“The main structural component of the fibrotic core is collagen and one of the most prominent collagens in the fibrotic kidney is collagen type III. C3M is a degradation product of collagen type III, produced by the matrix metalloproteinase (MMP)-9. C3M thereby reflects turnover of collagen type III in the interstitial matrix and can be considered as a marker for fibrotic activity[6].”

Revised Methods (page 5; lines 6-11): 

“Both serum and urine C3M were measured using competitive enzyme-linked immunosorbent assays (ELISAs) developed by Nordic Bioscience, Denmark [10, 15]. MMP-9 mediated degradation of collagen type III, produces a 10 amino acid neo-epitope (610’.KNGETGPQGP’619) (Fig 1). ELISAs were performed using two different monoclonal antibodies for detecting C3M in serum and urine, both antibodies specifically detected all fragments entailing the neo-epitope.”

- The definition of CKD progression was based on >30% decline of eGFR compared to baseline. It would be desirable that this >30% decline was based on repeated creatinine measurements, or is this already the case?

Thank you for this important comment. The decline in eGFR was based on repeated approximately annual measurements of creatinine, however the exact time from baseline to the creatinine measurements varied highly between the participants. 

This has been clarified in the Methods section (page 6; lines 12-15): “Approximately annual measurements of plasma creatinine were performed in 175 of the 198 participants (88.4%) as part of their regular diabetes control at Steno Diabetes Center Copenhagen. The exact time from baseline to each creatinine measurement varied highly between each individual and each measurement.”

-The uC3M story is not fully clear. uC3M is not mentioned in Table 1, but it is detected in urine as shown in Suppl figure 1. Since C3M is apparently a cleavage fragment and stable (since detected in serum) one would expect a circulating peptide to be filtered by the kidney and then found in urine. Do the authors have any explanation here?

Thank you for this question. In our study there is no correlation between the levels of C3M in serum and urine. One explanation for the lack of correlation may be that the C3M fragments can be of different lengths and that a large proportion of the circulating fragments in serum are not freely filtered. Another explanation for the lack of correlation between C3M in serum and urine may be that a large proportion of the fragments found in urine are due to degradation of collagen type III directly in the renal stroma, which are then released directly into the urine. The lack of correlation (or very poor correlation) between C3M in circulation and urine is a phenomenon observed across several other studies, and it is therefore not a random finding in the current investigation.

We have added these considerations in the Discussion section (page 15; lines 10-14): “We found no correlation between serum and urine C3M. This phenomenon has been observed across several other studies[6, 9]. One explanation could be that C3M fragments have different lengths and that a large proportion of the circulating fragments are not freely filtered into the urine. Another explanation could be that C3M produced by collagen type III degradation in the renal stoma were released directly into the urine and thereby not systemically available.”

- The last section of the results very briefly describes the interaction of sC3M with the inflammation markers. Since there is a quite strong correlation with those inflammatory markers and sC3M, what about the predictive value on CKD progression of CVD of those individual markers? It would be very interesting to compare the predictive value of those individual markers with sC3M. This is especially important since the authors claim that sC3M would be a potential marker of progression CKD and therefore its added value should be shown and compared to those inflammation markers.

Kindly see response to the comment below.

- Related to this, even if after adjustment with inflammation markers sC3M remained associated to outcome, is combination of one or more of those markers (logistic regression, SVMs, random Forrest, etc…) improving the predictive value of progression to CKD or CVD events over sC3M alone?

We agree with the Reviewer that this is relevant to investigate. We have compared the predictive value of sC3M with inflammation markers using ROC analyses. In the first model we included sC3M and the traditional cardiovascular risk factors, and in the second model we added the z-score of the inflammation markers. This did not change the AUC significantly for any of the defined endpoints. These findings are included in the revised: 

Revised Statistics (page 8; lines 1-6): “To evaluate the added predictive value for the markers of inflammation over the value of sC3M, we employed receiver operating characteristic (ROC) curves, applying C-statistics for area under curve (AUC) analysis. For reasons of statistical efficiency, we calculated a z-score for the markers of inflammation by averaging the individual biomarkers signed z-scores. The base model included the traditional cardiovascular risk factors and sC3M, and in the second model we added the z-score for the markers of inflammation.”

Results (page 14; lines 9-11): “Adding the z-score for the inflammation markers to ROC-curves with sC3M and traditional cardiovascular risk factors did not improve the predictive value on any of the endpoints (CVD events: p=0.36; all-cause mortality: p=0.32; >30% decline in eGFR: p=0.75).”

Discussion (page 16; lines 11-13): “The findings from our ROC analyses showed that adding the markers of inflammation to a model including traditional cardiovascular risk factors and sC3M did not add predictive value for any of the endpoints.”

- The strong association to “endothelial inflammation” of sC3M and the absence to predict CVD is surprising, this is not discussed.

We have, in accordance with this helpful comment, included the following to the revised Discussion (page 16; lines 3-11): “Our results showed consistent, positive correlations between sC3M and markers of inflammation and some of the markers of endothelial dysfunction. Supporting these findings, another study showed a positive correlation between sC3M and CRP in CKD [6]. These consistent correlations indicate that these pathways are interconnected in the pathogenesis of DKD. Despite the consistent and strong correlations, sC3M was not associated with development of CVD and mortality in this population, even though associations between the markers of inflammation and endothelial dysfunction and risk of both CVD and mortality have previously been shown in the same population. This adds evidence to the conclusion that sC3M is related to kidney disease but not CVD.”

Reviewer #2: 

Thank you for giving me the opportunity to review this research manuscript. The authors aimed to study the association of serum and urinary C3M (a matrix metalloproteinase-9-mediated degradation product of collagen type III) with progression of chronic kidney disease. They additionally examined the association of markers of systemic inflammation and endothelial dysfunction in relation to C3M. The authors documented that chronic kidney disease progression associates with high levels of serum C3M – urinary C3M was not associated. Please, find below some important statistical and methodological considerations:

Major comments:

- The median follow-up was six years and the creatinine was collected annually, which means patients would have an average of six creatinine measures. Considering this, using a categorical variable excludes important information about the progression of chronic kidney disease. From a clinical perspective, I understand that it makes more sense to show Cox proportional modeling because readers are more familiar with it. Nevertheless, the authors should include the rich information they have about the progression of kidney disease based on repeated creatinine measures. Therefore, I would strongly recommend they conduct and report mixed models. Just as an example, by running mixed modeling, authors could express the following estimates: kidney function decrease -0.2 mg/dL per 2-standard deviation (SD) in the sC3M. Moreover, if they estimate the difference change divided by follow-up time (progression rate), they could express the following: kidney function decrease -0.2 mg/dL per year per each 2-SD increment in the sC3M... It has a different interpretation (estimates) than Cox proportional models, but still, those estimates are clinically meaningful and from a statistical perspective, considering the outcome as a continuous repeated variable provides more accurate information than dividing into a categorical outcome.

Thank you for these important considerations. We agree that the decrease in eGFR over time as a continuous measure could provide useful clinical information. As recommended, we have analyzed the data using a mixed effects model. However, we have been challenged by the fact that the time from baseline to the approximately annual measurements of eGFR varied highly, both between individuals and between each eGFR measurements. Therefore, time had to be entered in the model as a continuous variable. In the analysis we demonstrate a significant association between sC3M and eGFR decline over time (P=0.037), but because of the continuous variables, the interpretation of the estimate is impossible. We have therefore chosen not to include the results in the manuscript. 

Revised Methods (page 7, lines 18-21): “Linear mixed effect models were applied to examine the association between sC3M/uC3M and eGFR decline during follow-up (including the five follow up measurements). The model was applied using the “gls function” with compound symmetry covariance structure in the “nlme package” using R version 4.1.0.” 

Revised results (page 13, line 23-24 to page 14 line 3): “The mixed model analysis showed a significant association between sC3M and eGFR decline over time in both the unadjusted (p = 0.037) and adjusted model (p = 0.048), including adjustment for sex, age, body mass index, LDL cholesterol, smoking, HbA1c, plasma creatinine, systolic blood pressure and urinary albumin excretion rate. Urine C3M was not associated with eGFR decline over time in neither the unadjusted (p = 0.24) or adjusted model (p= 0.36). ”

- I believe the authors should focus the manuscript on the progression of kidney disease and cardiovascular (CV) diseases as outcomes should be considered as competing risk. The CV outcomes are not part of the authors' hypothesis (which seems they are trying to test different hypotheses making the manuscript more difficult to digest). As they wrote in their introduction about why investigating CV outcomes, this is a pure statistical issue called "competing risk". As this is a predictive study, authors should treat CV disease and mortality as potential competing risks for the progression of kidney disease. That is, because these are high-risk patients, they can potentially die or suffer CV events (competing risk) before they develop the event (progression of kidney disease). Please, see the following paper which might help the authors to address their hypothesis and redefine their statistical approach: “Noordzij M. et al. (2013). When do we need competing risks methods for survival analysis in nephrology?. Nephrology Dialysis Transplantation, 28(11), 2670-2677.” Therefore, they should document the sub-hazard ratios for progression of kidney disease by adjusting models for total mortality and CV deaths as competing risk.

Thank you for this comment. We completely agree that competing risk of death should be considered and have included mortality as competing risk in the analysis for eGFR decline throughout the manuscript. Results were largely unchanged. 

Statistics (page 7; lines 11-15): “Cox proportional hazards analysis was used to calculate hazard ratios (HR) with 95% confidence intervals (CI) per doubling of sC3M and uC3M for the three endpoints and with mortality as competing risk in the analysis of eGFR decline. Adjustment for all models included traditional cardiovascular risk factors including sex, age, body mass index, LDL cholesterol, smoking, HbA1c, plasma creatinine, systolic blood pressure and UAER. 

Results (page 12; lines 12-15): “Table 3 shows the associations between sC3M, uC3M and the three endpoints. Higher sC3M was associated with an increased risk of a >30% decline in eGFR in the unadjusted and adjusted analyses, including mortality as competing risk.”

Table 3. Unadjusted and adjusted associations between serum and urine C3M and development of cardiovascular events, all-cause mortality and decline in eGFR

 Cardiovascular events

 (n = 40) All-cause mortality

 (n = 26) Decline in eGFR > 30% with death as competing risk 

(n=42)*

 HR (95% CI) P-value HR (95% CI) P-value HR (95% CI) P-value

Serum C3M 

Unadjusted 1.17 (0.57-2.40) 0.67 1.57 (0.67-3.71) 0.30 2.71 (1.38-5.30) 0.004

Adjusted 1.52 (0.67-3.42) 0.31 1.80 (0.69-4.73) 0.23 2.98 (1.41-6.26) 0.004

Urine C3M 

Unadjusted 0.76 (0.51-1.12) 0.16 0.88 (0.52-1.49) 0.21 0.94 (0.70-1.28) 0.70

Adjusted 0.83 (0.60-1.14) 0.25 0.93 (0.62-1.40) 0.74 0.96 (0.66-1.38) 0.82

- If the authors divided uC3M levels by urinary creatinine levels, and then estimated glomerular filtration rate (GRF) based on serum creatinine levels, should not the division affect the authors analysis? I understand the biological reasoning, but there is a statistical consideration about this. Can the authors perform the analysis for uC3M without this adjustment/standardization by serum creatinine levels?

Thank you for these considerations. We have included sensitivity analysis without the adjustment for serum-creatinine levels. Results were substantially unchanged. 

Results (page 14; line12 to page 15 line 2): “Urinary C3M were divided with urine creatinine to normalize for urine output. Since this is correlated with the serum creatinine level, we performed a sensitivity analysis not including serum-creatinine in the adjusted model for eGFR decline. This did not change the estimate or the lack of association with eGFR decline, HR (95% CI): 0.88 (0.63 – 1.25), p-value 0.47.”

- Figure 1, can the authors indicate the cutoff points for the t1, t2, and t3? Moreover, authors should standardize figure 1 by age (tertile groups, or two groups based on the mean age) and sex groups. Another alternative is to standardized the incidence curve for the average population of con founders (see example). Otherwise, these are crude analysis. Can the authors also provide the same curve for uC3M? I understand that it would not be significant, but the utility of Figure one does not only rely on reporting the absolute risk for progression of chronic kidney disease but also to facilitate readers quick results of the study.

Thank you, we have replaced the former Figure 1 with incidence curves adjusted for sex, age, body mass index, LDL cholesterol, smoking, HbA1c, plasma creatinine, systolic blood pressure and urinary albumin excretion rate. As suggested, we have provided the same curves for both sC3M and uC3M. The figure is now Fig 2. 

Statistics (page 7; lines 16-17): “Cox proportional hazard analysis was applied to compare the risk of eGFR decline according to tertiles of sC3M and uC3M, adjusted for traditional cardiovascular risk factors as described.”

Results (page 12; line 16 to page 17 line 2): “Fig 2 shows the Cox proportional hazard plots with tertiles of sC3M/uC3M and a >30% decline in eGFR. High levels of sC3M were associated with a significantly increased risk of decline in eGFR > 30% (p = 0.047). The risk was significantly higher in tertile 3 compared with tertile 1 (p =0.014). Tertile 2 was not significantly different from tertile 1 (p = 0.132). There was no association between tertiles of uC3M and development of a >30% decline in eGFR (p= 0.96).”

Minor comments:

- Can the authors specify how much is the SD doubling in the log-transformed C3M? 

Unfortunately, this was not correctly described. Both sC3M and uC3M were log2-transformed, but not standardized. The HR reflects a doubling of sC3M and uC3M. This has been corrected throughout the manuscript. 

- Table 2, can the authors provide the 95% CI? 

The confidence intervals have been added to Table 2. 

- There is no documentation of how urinary C3M was measured. Similar to sC3M? 

Urinary C3M was measured similar to serum C3M using ELISA. Another monoclonal antibody, but the antibodies recognize the same amino acid sequence. The Methods section has been revised. 

Revised Methods (page 5; lines 6-11): “Both serum and urine C3M were measured using competitive enzyme-linked immunosorbent assays (ELISAs) developed by Nordic Bioscience, Denmark [10, 15]. MMP-9 mediated degradation of collagen type III, produces a 10 amino acid neo-epitope (610’.KNGETGPQGP’619) (Fig 1). ELISAs were performed using two different monoclonal antibodies for detecting C3M in serum and urine, both antibodies specifically detected all fragments entailing the neo-epitope”

- Change title of table 3. 

We have changed the title to match the other table titles. 

- There is an extra decimal in the last estimate of table 3. 

Thank you, this has been changed accordingly. 

- Why not include BP treatment as confounder. 

We have added a sensitivity-analysis including BP treatment in the adjusted models. Results were substantially unchanged.

Results (page 15; lines 3-4): “Additional sensitivity analysis showed that adding antihypertensive treatment to the model of eGFR decline did not change the results, HR (95% CI): 2.94 (1.40-6.19).”

- Specify follow-up in time in the abstract. 

The follow-up time has been added to the abstract

- How many creatinine measures were collected? 

Creatinine was measured approximately annually during follow-up, which was five measurements. This information is included in the revised Methods (page 6; lines 12-15): “Annual measurements of plasma creatinine were performed in 175 of the 198 participants (88.4%) as part of their regular diabetes control at Steno Diabetes Center Copenhagen. The exact time from baseline to each creatinine measurement varied highly between each individual and each measurement.”

175 participants had complete follow-up with five creatinine measurements. 15 participants had four measurements, 11 had three measurements and 12 had two creatinine measurements. 

- Supplemental figure 1 can be placed in the manuscript. I believe it is a main (additional) finding that sC3M and uC3M were not correlated. Moreover, it is just one item as supplemental information [it can also be described in the results section as: “sC3M and uC3M were not correlated (P value >x.xxx, data not shown) ….]. On the other hand, a potential supplemental table is to replicate table 1 but for uC3M.

Yes, as suggested, we have added a table with the baseline characteristics for tertiles of uC3M to the supplemental information. We have therefore chosen to keep the figure with correlations between sC3M and uC3M in the supplement.

---

## [Decision Letter · Decision Letter 1]

6 Mar 2023

Marker for kidney fibrosis is associated with inflammation and deterioration of kidney function in people with type 2 diabetes and microalbuminuria

PONE-D-22-32292R1

Dear Dr. Poulsen,

We’re pleased to inform you that your manuscript has been judged scientifically suitable for publication and will be formally accepted for publication once it meets all outstanding technical requirements.

Kind regards,

Harald Mischak

Academic Editor

PLOS ONE

Additional Editor Comments (optional):

Reviewers' comments:

Reviewer's Responses to Questions

**Comments to the Author**

1. If the authors have adequately addressed your comments raised in a previous round of review and you feel that this manuscript is now acceptable for publication, you may indicate that here to bypass the “Comments to the Author” section, enter your conflict of interest statement in the “Confidential to Editor” section, and submit your "Accept" recommendation.

Reviewer #1: All comments have been addressed

2. Is the manuscript technically sound, and do the data support the conclusions?

Reviewer #1: Yes

3. Has the statistical analysis been performed appropriately and rigorously? 

Reviewer #1: Yes

4. Have the authors made all data underlying the findings in their manuscript fully available?

Reviewer #1: Yes

5. Is the manuscript presented in an intelligible fashion and written in standard English?

Reviewer #1: Yes

6. Review Comments to the Author

Reviewer #1: I would like to thank the authors for the excellent and clear revision. I do not have further comments on the manuscript.

7. PLOS authors have the option to publish the peer review history of their article (what does this mean?). If published, this will include your full peer review and any attached files.

Reviewer #1: No

---

## [Editor Report · Acceptance letter]

9 Mar 2023

PONE-D-22-32292R1 

Marker for kidney fibrosis is associated with inflammation and deterioration of kidney function in people with type 2 diabetes and microalbuminuria 

Dear Dr. Poulsen:

I'm pleased to inform you that your manuscript has been deemed suitable for publication in PLOS ONE. Congratulations! Your manuscript is now with our production department. 

Kind regards, 

on behalf of

Prof. Harald Mischak 

Academic Editor

PLOS ONE